**Data Availability Statement:** All relevant data are within the manuscript and its Supporting Information files.

# Does menstrual hygiene management and water, sanitation, and hygiene predict reproductive tract infections among reproductive women in urban areas in Ethiopia?

**Ayechew Ademas[1]⦿, Metadel Adane**  **[1]⦿\*, Tadesse Sisay[1], Helmut Kloos[2], Betelhiem Eneyew[1], Awoke Keleb[1], Mistir Lingerew[1], Atimen Derso[1], Kassahun Alemu[3]**

**1** Department of Environmental Health, College of Medicine and Health Sciences, Wollo University, Dessie, Ethiopia, **2** Department of Epidemiology and Biostatistics, University of California, San Francisco, CA, United States of America, **3** Department of Epidemiology and Biostatistics, Institute of Public Health, University of Gondar, Gondar, Ethiopia

⦿ These authors contributed equally to this work.

\* metadel.adane2@gmail.com

## Abstract

Reproductive tract infections (RTIs) are a public health concern in Ethiopia. However, the relationship between menstrual hygiene management (MHM) and water, sanitation, and hygiene (WASH) factors to RTIs have not been well addressed. A community-based cross-sectional study was conducted from January to March 2019 among 602 systematically selected reproductive-age women aged 15–49 years in Dessie City. Data were collected using a questionnaire and a direct observation checklist. RTIs were identified by the presence during one year before data collection of one or more signs of vaginal discharge, itching/irritation or ulcers/lesions around the vulva, pain during urination and sexual intercourse, and lower abdominal pain and lower back pain. Data were analyzed using multivariable logistic regression analysis with 95%CI (confidence interval). The self-reported prevalence of RTIs was 11.0%(95%CI:8.5–13.7%) during one year prior to the survey. The most commonly reported symptoms of RTI were burning micturition (9.1%) and vaginal discharge (6.1%). Three-fourths 443(75.0%) of households used traditional pit latrines and the majority of the study participants 527(89.2%) did not meet the basic access requirement of 20 liters of water per capita per day. The majority 562(95.1%) of the study participants did not have multiple sexual partners during the last year and 97.8% did not practiced sexual intercourse during menstruation. The most common type of blood-absorbent material used was a sanitary pad 497(84.8%) followed by cloth 89(15.2%). Factors significantly associated with RTIs were using unclean latrines (AOR: 4.20; 95%CI:2.00–8.80), not washing hands with soap before touching the genital area (AOR: 3.94; 95%CI:1.49–10.45), history of symptoms of RTIs in the past year (AOR: 5.88; 95%CI:2.30–14.98), having multiple sexual partners in the past year (AOR: 4.46; 95%CI:1.59–12.53), changing absorbent material only once per day (AOR: 8.99; 95%CI:4.51–17.92), and washing the genital area only once per day during

**Funding:** Wollo University Funded this study. The funders had no role in study design, data collection and analysis, decision to publish, or preparation of the manuscript.

**Competing interests:** The authors have declared that no competing interests exist.

**Abbreviations:** AOR, adjusted odds ratio; CI, confidence interval; COR, crude odds ratio; MHM, menstruation hygiene management; RTI, reproductive tract infection; STI, sexually transmitted infection; WASH, water, sanitation and hygiene.

menstruation (AOR: 5.76; 95%CI:2.07–16.05). The self-reported prevalence of RTI showed that one women experienced RTI among ten reproductive-age women. Designing a women's health policy that focuses on ensuring availability of WASH facilities and improving MHM at the community level is key for sustainably preventing RTIs.

## Introduction

Reproductive tract infections (RTIs) are infections of the reproductive system and belong to a group of communicable diseases that are transmitted predominantly by sexual contact (mainly chlamydia, gonorrhea, chancroid, syphilis and HIV) but also include endogenous infections and iatropenic infections linked with unsafe abortions and poor delivery practices [1]. A woman with an RTI can experience various symptoms ranging from simple backache to lower abdominal pain, genital ulcers, vulvar itching, inguinal swelling, abnormal vaginal discharge, burning sensation, and dyspareunia [2].

RTIs are among the most common causes of illness among women of reproductive age worldwide and especially in developing countries. Nevertheless, public health research, interventions, and services give very little attention to RTIs [3]. The National Family Health Survey of India reported that 39.2% of women in India surveyed between 2006 and 2008 had one or more diagnosed RTIs whereas the prevalence of self-reported RTI symptoms was 11–18% in various nationally representative studies [4]. In India alone, healthcare workers estimate about 40 million new cases of RTI emerge each year [5].

RTIs are highly prevalent in Ethiopia. For instance, in a study of 210 patients conducted at St. Paul's Hospital in Addis Ababa between September 2015 and July 2016, the overall prevalence of bacterial vaginosis was reported at 48.6% [6]. Surveillance in Ethiopia from July 2014 to June 2015 revealed a total of 1,421 sexually transmitted infection (STI) cases from 20 sentinel surveillance sites; the 1,509 STI symptom episodes were mostly vaginal discharge (52.2%), followed by urethral discharge (25.3%), lower abdominal pain (13.3%), and genital ulcers (7.4%). Of the total number of cases, 968 (68.1%) were females, who bore the highest burden [7].

The impact of menstrual hygiene, which is critical for women, has been largely neglected by water, sanitation, and hygiene (WASH) sector researchers. As a result, millions of women and girls continue to be denied WASH, health, education, dignity, and gender equity [8]. In Gujarat in India, 91% of girls reported staying away from flowing water during menstruation. In a study in South Asia, 20% of the women who had access to toilets stated that they refrained from using them during their periods, partly due to fear of staining the toilet. Menstruation is a natural process, but if not properly managed it can result in health problems. Researchers have suggested links between unhygienic menstrual hygiene management and urinary infections, RTIs, and other diseases [9].

If RTIs are left untreated or treatment is delayed, complications can result such as pelvic inflammatory disease, infertility, cervical cancer, puerperal sepsis, chronic pelvic pain, ectopic pregnancy, pregnancy loss, preterm delivery, neonatal blindness, premature membrane rupture, and low birth weight [10]. There is a paucity of relevant data on the overall prevalence of RTIs in community-based settings in Ethiopia, and the association between RTIs and menstruation hygiene management (MHM) and WASH factors has not been established. Therefore, this study aimed examined the prevalence of RTIs and associated factors among reproductive-age women in Dessie City, Ethiopia. The findings may contribute information towards narrowing this gap in the research.

## Materials and methods

### Study design and description of the study area

A community-based cross-sectional study was conducted in Dessie City from January to March 2019. Dessie is the capital of South Wollo Zone, located about 400 km from Addis Ababa, the capital of Ethiopia. The city is located at an altitude of 2,470 to 2,550 meters. The Dessie City Administrative Health Office estimated the population of Dessie in 2018/2019 was 223,639, of which 104,800 (46.86%) were males and 118,839 (53.14%) were females [11].

There are 16 *kebeles* in Dessie City, 10 urban and 6 rural. A *kebele* is the smallest administrative unit in Ethiopia, consisting of about 5,000 people. This study was limited to three randomly selected urban *kebeles* (*kebeles* 02 [Salayish], 07 [Dawido], and 09 [Robit]). In 2018/2019, the city contained 52,009 households, 42,745 (82.19%) urban and 9264 (17.81%) rural. Of the 52,736 women in the reproductive-age group, 43,342 (82.2%) lived in the urban and 9,394 (17.8%) in the rural parts of Dessie City [11].

### Source population, inclusion and exclusion criteria

The source population consisted of all women of reproductive age in the urban part of Dessie City. Women aged 15–49 years who menstruated during the three months prior to data collection were included. Women aged 15–49 who were pregnant and/or who had not menstruated in the three months prior to data collection were excluded.

### Sample size determination

A single proportion sample size estimation formula [12] was used with the following assumptions:

$$n = \frac{(z_{a/2})^2 * p(1 - p)}{d^2}$$

$Z_{\alpha/2}$ is the standard normal variable value at 95% CI (confidence interval) (α is 0.05 with 95%CI, $Z_{\alpha/2}$ = 1.96; an estimate of the prevalence (*P*) of RTI 50%, and 5% the margin of error (d). A design effect of 1.5 was used due to multistage sampling and a sample size correction formula was employed since the source population was less than 10,000 in the study area. A 10% non-response rate was used to obtain 602 as an adequate sample size.

### Sampling technique and procedures

A two-stage sampling technique was used to select women of reproductive age. First, three *kebeles* were selected randomly out of ten *kebeles*. A sampling frame was prepared for a house-to-house survey of households in the selected three *kebeles* (*kebeles* 02, 07, and 09) by identifying a women of reproductive age. We found a total of 8,845 households containing 8,969 reproductive-age women. Proportion-to-size allocation was made to determine the sample size required from each selected *kebele*.

The sampling units were households having at least one woman of reproductive age. In the second stage of sampling, systematic random sampling with a fixed interval of 14 was used to select sampling unit households; the simple random lottery method was used for selecting the first household. When more than one reproductive-age women was present in a selected household, the lottery method was used to select the study participant.

## Outcome measurement

The outcome variable of this study was RTI presence (yes) or absence (no) during the one year prior to data collection. Measurement of RTI was based on syndromic management of RTI, a method recommended by the World Health Organization for low-income and low-resource settings such as Ethiopia. Thus, the presence of RIT was presumed if one or more of the following signs occurred: vaginal discharge, itching/irritation or ulcers/lesions around the vulva, pain during urination and sexual intercourse, and lower abdominal and lower back pain [2].

## Operational definitions

**Menstruation hygiene management (MHM).** Use of clean material to absorb or collect blood that can be changed in privacy as often as necessary for the duration of the menstruation period, use of soap and water for washing the body as required, and access to facilities for disposing of used menstrual management material [13]. MHM was assessed for the three months immediately prior to data collection.

**History of RTI symptoms.** The presence or absence of history of RTI was considered for the one-year period prior to three months before data collection.

**Ulcers/lesions around vulva.** Vesicular/non-vesicular, recurrent/non-recurrent, multiple/solitary ulcer on the labia, vagina, or rectum, with or without inguinal lymphadenopathy [2]. The presence or absence of ulcers/lesions around the vulva was assessed for the year prior to data collection.

**Vaginal discharge syndrome.** A change in the amount, color, and odor of the vaginal discharge [2]. The presence or absence of vaginal discharge syndrome for this study was assessed for the year prior to data collection.

**Lower abdominal pain syndrome.** Bilateral lower abdominal or pelvic pain or lower abdominal tenderness together with cervical excitation tenderness, or a tender pelvic mass together with fever, nausea, or vomiting [2] was assessed for the year prior to data collection.

**Clean latrine.** No fecal matter observed in or around the pit latrine, latrine properly swept during the data collection period [14].

**Latrine utilization.** Presence of functional latrine, safe disposal of child feces, no observable feces in the compound, and presence of at least one sign of latrine use (footpath to the latrine not covered by grass, the latrine is smelly, absence of spider webs in the squatting hole, presence of anal cleansing material, fresh feces in the squatting hole, and the slab is wet) [15].

**WASH.** The collective term for water, sanitation, and hygiene. Due to their interdependence, these three core issues are grouped together in this study. WASH variables used in this study have been defined in other studies [16–18].

## Data collection procedures and quality assurance

The data collection tool was a structured questionnaire adapted from similar studies. The questionnaire comprised socio-demographic variables of the study participants, WASH variables of the household, history of co-morbidity, behavioral variables, MHM practices of participants, and the outcome variable.

The questionnaire, originally developed in English, was translated into Amharic (local language) and back into English by language experts to ensure consistency. Data were collected through face-to-face interviews and using a direct observation checklist. Six female midwives with BSc degrees and two female environmental health professionals with BSc degrees were employed as data collectors and supervisors, respectively.

Trained female midwives were employed to improve communication between study participants and data collectors and ensure the acquisition of reliable data about RITs and other

independent variables. Data collectors and supervisors were trained by the principal investigator for two days on the objectives of the study, the content of the questionnaire, ethical issues, and approaches to be used during data collection.

Inter-observer reliability was ensured by providing clear definitions of WASH and events to be recorded, by training data collectors, and by providing feedback about discrepancies during daily supervisions. We re-interviewed 5% of the study participants, using a different interviewer, to check reliability of the information entered by different interviewers. The qualifications of the interviewers and the training they received reduced the likelihood for interviewer bias. We also pre-tested the questionnaire with 30 women of reproductive age (5% of the sample size) in one non-selected *kebele* (*kebele* 03 in Dessie City) before the actual data collection and made changes to the questionnaire where necessary for clarity. During administration of the survey, the collected data were checked daily by the principal investigator and supervisors for completeness, and houses providing incomplete data were revisited once to obtain further data.

### Data management and analysis

The data were coded, entered into EpiData Version 3.1, and exported to SPSS (Statistical Package for the Social Sciences) Version 25.0 statistical software for analysis. Descriptive statistics such as frequency distribution and prevalence were computed. Principal component analysis was done to construct the household wealth index with the following considerations: communality value > 0.5, Kaiser-Meyer-Olkin (KMO) value > 0.5, and eigen values greater than one [19]. Multicollinearity between independent variables was checked with the standard error of the coefficient with a cut-off point greater than 2 [20], which was not observed.

Associations between independent variables and RTI were determined using a binary logistic regression model. First, the bi-variate analysis was done and all independent variables having $P$-values $\leq 0.25$ were retained for multivariable analysis. $P$-value $< 0.05$ and adjusted odds ratio (AOR) with 95%CI were considered as statistically significant and used to determine factors associated with RTI in the final model. Model fitness was checked using the Hosmer and Lemeshow test [20] to run logistic regression analysis with cut-off point $P$-value $> 0.05$.

### Ethics approval and consent to participate

Ethical clearance was obtained from the Ethical Review Committee of the College of Medicine and Health Sciences, Wollo University. After the purpose of the study was explained and before data collection, oral/verbal informed consent was obtained from the study participants. For study participants below 18 years of age, verbal consent was obtained from their parents/guardians. Participants' privacy and the confidentiality of the information they gave were ensured at all levels; the interviews were conducted in a private place to ensure privacy and personal identifiers were not used. Participants who reported symptoms of RTI at the time of data collection were advised to obtain treatment at a health facility and necessary arrangements were made for them to do so with health extension workers.

## Results

### Socio-demographic and economic characteristics

Data were collected from 602 reproductive-age women, 591 of whom completed the process (response rate of 98.17%). Of the study participants, 318(53.8%) were Muslims and 258 (43.7%) were Orthodox Christians. Their ages ranged from 16 to 49 years with a median of 30 and interquartile range of 10. Only 388(65.65%) were married. About one-fourth of the study

participants were found in the highest 117(19.8%) and lower 116(19.63) economic classes (Table 1).

## Prevalence of reproductive tract infections

The prevalence of RTI among reproductive-age women was 11%(95%CI:8.5–13.7%). Among symptomatic women, the symptoms most commonly reported were burning micturition (9.1%) and vaginal discharge (6.1%). Lower abdominal pain (1.4%) and lower back pain (0.8%) were the least frequently reported symptoms (Fig 1).

## Water, sanitation, and hygiene-related factors

A latrine was present in all study households; 443(75.0%) households had traditional pit latrines, 78(13.2%) had pour-flush latrines, and 70(11.8%) had ventilated improved pit latrines.

**Table 1.  Socio-demographic and economic characteristics of the study participants in Dessie City, Ethiopia, January to March 2019.**

| Variables | Frequency | RTI | | COR (95%CI) | P-value |
|---|---|---|---|---|---|
| | | Yes | No | | |
| | n(%) | n | n | | |
| **Age in years** | | | | | |
| 15–19 | 42(7.1) | 10 | 32 | 2.96(1.24–7.05) | 0.014 |
| 20–34 | 371(62.8) | 38 | 333 | 1.08(0.59–1.97) | 0.800 |
| 35–49 | 178(30.1) | 17 | 161 | 1 | |
| Marital status | | | | | |
| Single | 133(22.5) | 29 | 104 | 2.42(0.68–8.55) | 0.171 |
| Married | 388(65.7) | 23 | 365 | 0.55(0.15–1.94) | 0.349 |
| Divorced | 41(6.9) | 10 | 31 | 2.80(0.70–11.24) | 0.148 |
| Widowed | 29(4.9) | 3 | 26 | 1 | |
| Religion | | | | | |
| Orthodox Christianity | 258(43.7) | 23 | 235 | 0.19(0.14–2.99) | 0.567 |
| Muslim | 318(53.8) | 37 | 281 | 0.26(0.20–4.30) | 0.931 |
| Protestant | 15(2.5) | 5 | 10 | 1 | |
| Educational status* | | | | | |
| Illiterate | 70(11.8) | 7 | 63 | 1.17(0.42–3.22) | 0.766 |
| Primary | 160(27.1) | 20 | 140 | 1.50(0.67–3.34) | 0.321 |
| Secondary | 246(41.6) | 28 | 218 | 1.35(0.63–2.88) | 0.440 |
| Above secondary | 115(19.5) | 10 | 105 | 1 | |
| Number of parity | | | | | |
| Three to five | 196(33.2) | 19 | 177 | 0.36 (0.19–0.69) | 0.002 |
| One to two | 277(46.9) | 19 | 258 | 0.25(0.13–0.47) | < 0.001 |
| None | 118(19.9) | 27 | 91 | 1 | |
| Wealth index | | | | | |
| Lower | 116(19.6) | 8 | 108 | 0.34(0.14–0.80) | 0.014 |
| Second | 120(20.3) | 6 | 114 | 0.24(0.09–0.62) | 0.003 |
| Middle | 119(20.1) | 13 | 106 | 0.56(0.27–1.18) | 0.128 |
| Fourth | 119(20.1) | 17 | 102 | 0.76(0.38–1.53) | 0.445 |
| Highest | 117(19.8) | 21 | 96 | 1 | |

[1]reference category.

*illiterate not attended both formal and informal education, primary education means Grade 1–8; secondary education Grade 9–12, and above secondary education means that diploma, degree, masters and others.

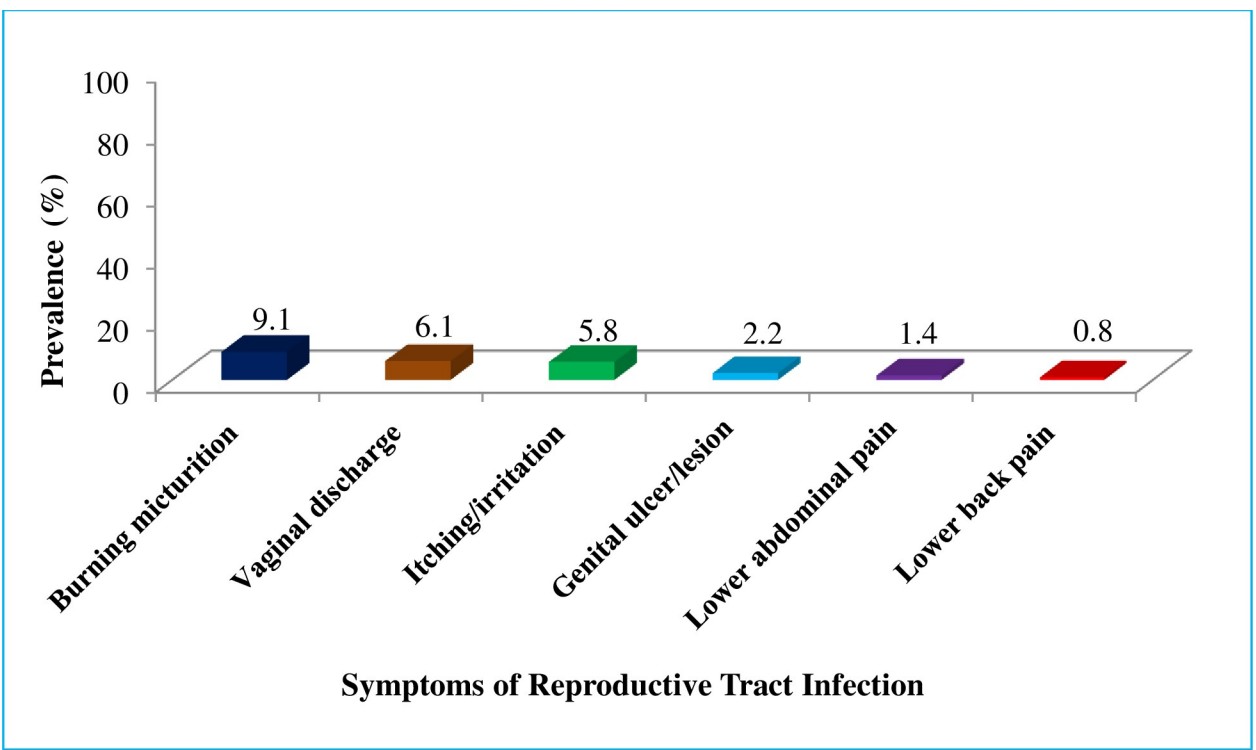

**Fig 1. Self-reported symptom-based prevalence of RTI among reproductive-age women in Dessie City, northeastern Ethiopia, January to March 2019.**

The distance between latrine and house ranged from 0 to 30m with a median of 6m and an interquartile range of 5m. The majority of the respondents 527(89.2%) did not meet the basic access requirement of 20 liters water per capita per day. The amount of water consumed per capita per day ranged from 3 to 30 liters with a median of 10 and an interquartile range of 7 (Table 2).

### History of co-morbidities

Of the total respondents, 38(6.4%) reported at least one symptom of RTI in the year prior to three months before data collection. Sixty-two (10.5%) of the study participants reported having had an abortion in the year prior to data collection (Table 3).

### Behavioral factors

The majority (95.1%) of the study participants did not have multiple sexual partners during the previous year and 97.8% practiced sexual intercourse during menstruation. The bivariate analysis showed that those respondents who had multiple sexual partners in the year before data collection were at a 12.89 times higher risk for developing RTI (COR: 12.89; 95%CI:5.86–28.34) compared to those who did not have multiple partners (Table 4).

### Menstrual hygiene management practices

The majority of respondents 586(99.2%) used blood-absorbent material during menstruation. The most common type of blood-absorbent material used was a sanitary pad 497(84.8%) followed by cloth 89(15.2%). Of the study subjects who used blood-absorbent material, 537

**Table 2. Water, sanitation, and hygiene-related characteristics among the study participants in Dessie City, Ethiopia, January to March 2019.**

| Variables | Frequency | RTI | | COR (95%CI) | P-value |
|---|---|---|---|---|---|
| | | Yes | No | | |
| | n(%) | n | n | | |
| Cleanliness of latrine used | | | | | |
| Not clean | 202(34.2) | 46 | 156 | 5.74(3.26–10.12) | < 0.001 |
| Clean | 389(65.8) | 19 | 370 | 1 | |
| Distance of latrine from the house | | | | | |
| < 15 meters | 547(92.5) | 60 | 487 | 0.96(0.77–42.09) | 0.088 |
| 15–30 meters | 44(74.5) | 5 | 39 | 1 | |
| Amount of water used per capita per day | | | | | |
| <20 litters | 527(89.2) | 60 | 467 | 1.51(1.19–23.88) | 0.048 |
| ≥20 litters | 64(10.8) | 5 | 59 | 1 | |
| Availability of water near latrine/toilet for hand washing | | | | | |
| No | 498(84.3) | 63 | 435 | 6.59(1.58–27.42) | 0.010 |
| Yes | 93(15.7) | 2 | 91 | 1 | |
| Water source | | | | | |
| Public stand pipe | 12(2.0) | 5 | 7 | 5.85(0.35–7.65) | |
| House/yard connected to tap water | 579(98.0) | 63 | 516 | 1 | 0.530 |
| Hand washing with soap before touching the genital area during menstruation | | | | | |
| No | 390(66.0) | 59 | 331 | 5.79(2.46–13.67) | < 0.001 |
| Yes | 201(34.0) | 6 | 195 | 1 | |

[1]reference category.

(91.6%) changed their blood-absorbent material every day and 462(78.8%) changed the blood-absorbent material two or more times per day during menstruation (Table 5).

## Multivariable logistic regression analysis

In the multivariable analysis, socio-demographic factors were not significantly associated with RTI. Of the WASH factors, using unclean latrines (AOR: 4.20; 95%CI:2.00–8.80) and not washing hands with soap before touching the genital area (AOR: 3.94; 95%CI:1.49–10.45) were significantly associated with RTI. Of the history of co-morbidity factors, only having history of RTI symptoms in the year preceding three months before data collection was significantly associated with RTI (AOR: 5.88; 95%CI:2.30–14.98). Regarding behavioral factors, having

**Table 3. History of comorbidities among reproductive age women in Dessie City, Ethiopia, January to March 2019.**

| Variables | Frequency | RTI | | COR (95%CI) | P-value |
|---|---|---|---|---|---|
| | | Yes | No | | |
| | n(%) | n | n | | |
| History of abortion in year prior to data collection | | | | | |
| Yes | 62(10.5) | 12 | 50 | 2.16(1.08–4.30) | 0.029 |
| No | 529(89.5) | 53 | 476 | 1 | |
| History of RTI symptoms in year prior to three months before data collection | | | | | |
| Yes | 38(6.4) | 18 | 20 | 9.69(4.80–19.58) | < 0.001 |
| No | 553(93.6) | 47 | 506 | 1 | |

[1]reference category.

**Table 4.  Characteristics of behavioral factors among reproductive age women in Dessie City, Ethiopia, January to March 2019.**

| Variables | Frequency | RTI | | COR (95%CI) | P-value |
|---|---|---|---|---|---|
| | | Yes | No | | |
| | n(%) | n | n | | |
| Multiple sexual partners during the last year | | | | | |
| Yes | 29(4.9) | 16 | 13 | 12.89 (5.86–28.34) | < 0.001* |
| No | 562(95.1) | 49 | 513 | 1 | |
| Sexual intercourse during menstruation | | | | | |
| Yes | 13(2.2) | 4 | 9 | 3.77(1.13–12.60) | 0.031* |
| No | 578(97.8) | 61 | 517 | 1 | |
| Use of intrauterine contraceptive device (IUCD) | | | | | |
| Yes | 17(2.9) | 2 | 15 | 1.08(0.24–4.84) | 0.918 |
| No | 574(97.1) | 63 | 511 | 1 | |

[1]reference category.

**Table 5.  Menstrual hygiene management (MHM) practices among reproductive age women in Dessie City, Ethiopia, January to March 2019.**

| Variables | Frequency | RTI | | COR (95%CI) | P-value |
|---|---|---|---|---|---|
| | | Yes | No | | |
| | n(%) | n | n | | |
| Nature of absorbent material used during menstruation | | | | | |
| Cloth | 89(15.2) | 9 | 80 | 0.89(0.42–1.86) | 0.749 |
| Sanitary pad/napki | 497(84.8) | 56 | 441 | 1 | |
| Where do you put your menstruation pad/cloth | | | | | |
| I reuse it | 68(11.6) | 3 | 65 | 0.34(0.10–1.11) | 0.075 |
| I dispose of it | 518(88.4) | 62 | 456 | 1 | |
| Frequency of changing absorbent material during menstruation | | | | | |
| Alternate days | 49(8.4) | 18 | 31 | 6.05(3.15–11.64) | <0.001 |
| Every day | 537(91.6) | 47 | 490 | 1 | |
| Frequency of changing absorbent material per day during menstruation | | | | | |
| Once and below | 124(21.2) | 44 | 80 | 11.55(6.52–20.46) | <0.001 |
| Twice or more | 462(78.8) | 21 | 441 | 1 | |
| Did you take all body bath during menstruation | | | | | |
| No | 467(79.0) | 58 | 409 | 2.37(1.05–5.33) | 0.037 |
| Yes | 124(21.0) | 7 | 117 | 1 | |
| Days of taking all body bath during menstruation | | | | | |
| Not washing every day | 477(80.7) | 59 | 418 | 2.54(1.07–6.04) | 0.035 |
| Washing every day | 114(19.3) | 6 | 108 | 1 | |
| Do you wash the genital area every day during menstruation | | | | | |
| No | 62(10.5) | 25 | 37 | 8.26(4.53–15.07) | < 0.001 |
| Yes | 529(89.5) | 40 | 489 | 1 | |
| Frequency of washing the genital area per day during menstruation | | | | | |
| Once | 33(5.6) | 13 | 20 | 6.33(2.98–13.45) | < 0.001 |
| Twice or more | 558(94.4) | 52 | 506 | 1 | |
| What do you use to wash the genital area | | | | | |
| Water only | 499(94.3) | 35 | 464 | 1.88(0.33–18.54) | 0.383 |
| Water and soap | 30(5.7) | 5 | 25 | 1 | |

**Table 6. Factors associated with RTIs among reproductive age women from multivariable logistic regression analysis.**

| Variable | COR (95%CI) | AOR (95%CI) |
|---|---|---|
| Cleanliness of latrine used | | |
| Not clean | 5.74(3.26–10.12) | 4.20(2.00–8.80) |
| Clean | 1 | 1 |
| Hand washing with soap before touching the genital area | | |
| No | 5.79(2.46–13.67) | 3.94(1.49–10.45) |
| Yes | 1 | 1 |
| History of RTI symptoms in year prior to three months before data collection | | |
| Yes | 9.69(4.80–19.58) | 5.88(2.30–14.98) |
| No | 1 | 1 |
| Having multiple sexual partners in the past year | | |
| Yes | 12.89(5.86–28.34) | 4.46(1.59–12.53) |
| No | 1 | 1 |
| Frequency of changing absorbent material per day during menstruation | | |
| Once | 11.55(6.52–20.46) | 8.99(4.51–17.92) |
| Twice and above | 1 | 1 |
| Frequency of washing the genital area per day during menstruation | | |
| Once | 6.33(2.98–13.45) | 5.76(2.07–16.05) |
| Twice or more | 1 | 1 |

[1]reference category.

multiple sexual partners before the study was significantly associated with RTI (AOR: 4.46; 95%CI:1.59–12.53) (Table 6).

Regarding factors related to MHM, the odds of reproductive-age women developing RTI were 8.99 times higher (AOR: 8.99; 95%CI:4.51–17.92) in those who changed absorbent material only once per day than in those who changed absorbent material two or more times per day during menstruation. The odds of developing RTI were 5.76 times higher (AOR: 5.76; 95% CI:2.07–16.05) for women who washed the genital area once per day than for women who washed the genital area two or more times per day during menstruation (Table 6).

## Discussion

Our study used a community-based cross-sectional design to estimate the self-reported prevalence of RTI and associated factors among reproductive-age women (15–49 years) in Dessie City. Results show the prevalence of RTI during the year prior to the survey was 11.0%(95% CI:8.5–13.7%). We found that using unclean latrines, not washing hands with soap before touching the genital area, having had symptoms of RTI in the last year, having multiple sexual partners in the previous year, changing blood-absorbent material only once per day, and washing the genital area only once per day during menstruation were significantly associated with RTI.

Our finding of the 11% prevalence of RTI was similar to that of a study in Kerala, India (11.8%) [21]. However, several studies in other geographic areas with similar populations have reported a higher prevalence of RTI than the figure in our study. For instance, studies placed the prevalence of RTI in Mohali, Punjab, at 45% [22]; in Bangalore City at 29.15% [23]; in a rural area of Tamil Nadu at 55.5% [24]; in a rural area of Surendranagar District at 56.5% [25]; in an urban slum in Bidar, Karnataka, at 36.1% [26]; in Etawah District, Uttar Pradesh, at

46.76% [27]; and in an urban slum of northeast Delhi, India, at 43.9% [28]. The relatively low prevalence of RTI in our study might be due to the implementation of adolescent and reproductive health and STI prevention and control programs in urban areas of Ethiopia.

In this study, the type of blood-absorbent material used during menstruation was not significantly associated with the development of RTI. A similar study conducted in an urban slum of northeast Delhi, India, found prevalence of RTI significantly associated with using cloth rather than sanitary napkins during menses [28]. A worldwide systematic review study in different African countries found between 18% and 44% of women using menstrual pads [29]. The spread of covid-19 has been associated with increasing unavailability of sanitary napkins for many African women due to hording, calling for regulation of their distribution at the retail level [30].

Our study indicates that frequent changing of blood-absorbent material daily during menstruation was protective against the development of RTI. Similarly, in a study conducted in urban slums in India, participants who changed sanitary pads/cloths frequently and maintained good hygiene were at lower risk of developing RTIs [31]. Das *et al.*[32] reported that Indian women who used disposable absorbent pads were less likely to develop urogenital infections tan women using reusable pads, a relationship we did not examine in our study.

In this study, use of unclean latrines was associated with RTI development. The literature reports that poor sanitation may facilitate transmission where genital contact is made with infected genital fluids on the latrine toilet seat [33]. A review of the relationship between WASH and maternal and reproductive health indicated that health outcomes are broad and overlapping and that more research is needed to assess the effects of individual WASH exposures [34].

Our study also demonstrated that not washing hands with soap before touching the genital area was significantly associated with RTI and a significant predictor of RTI. Adequate hand washing with soap significantly reduces microorganism contamination of hands, which is a potential source of microbes in the vaginal area [35]. A study in rural Kwa-Zulu Natal reported that genital touching of women was also associated with bacterial vaginosis but not with chlamydia, gonorrhoea, syphilis, trichomoniasis, and herpes simplex virus-2 [36]. The authors attributed this result to increased vaginal PH resulting from use of soap that enhances lactobacilli survival in the vagina. Another possible reason may be introduction of microbes into the vagina along with ash or soil.

Consistent with our findings, a study in Odisha, India, found that RTI symptoms were more common among women who did not wash their hands versus women who washed their hands with soap or ash/soil/mud after defecation [37]. In our study in Addis Ababa slums, only 17.1% mothers of under-five children washed their hands using soap after defecation [18].

Our study also indicated that frequent washing of the genital area each day during menstruation is protective against the development of RTI among reproductive-age women. A study conducted in Dehradun, India, reported an association between RTI and poor menstrual hygiene as measured by washing the genitalia less often than twice per day during menstruation [38]. We also found that a history of RTI symptoms in the year before data collection was significantly associated with the development of RTI. A similar study in an urban training health center of a tertiary care hospital in India showed that RTI was significantly associated with a history of RTIs in the previous year [21].

History of abortion in the year prior to data collection was not found to be significantly associated with risk of RTI in our study. However, a similar study among women in urban slums of India showed that participants who had a history of abortion had higher odds of RTI symptoms than those who had no history of abortion [31]. The difference in our finding might

be due to the implementation of clean, safe, and standard procedures during an abortion. Inadequate information on reproductive health impacts of abortion and the persisting high rate of unintended pregnancies (121 million annually between 2015 and 2019) ending in abortion worldwide indicates the need for further research [39].

One of the limitations of this study is that no diagnostic tests were used to validate the self-reported prevalence of RTIs; our estimates were based on syndromic management of curable sexually transmitted infections and other RTIs, a method recommended by the World Health Organization for low-resource settings [2, 40]. However, in the absence of a laboratory diagnostic to accompany the analysis reported, this work may greatly underestimate or overestimate RTI prevalence and the true burden of RTIs. Another limitation of this cross-sectional study is the difficulty of establishing causal relationships between the dependent and independent variables. Furthermore, the results of this study may not be representative of the occurrence and underlying factors of RTIs at the national level because it was conducted only in Dessie City.

One strength of this paper is that WASH variables (presence of latrine, cleanliness of latrine, utilization of latrine, and distance of latrine from house) were observed and/or measured by data collectors to minimize self-reporting bias. Another strength is the fact that, unlike many previous studies of WASH interventions that focused on evaluating the impact of the variables on infectious diseases in children, this study examined gender-specific outcome measures of RTI at the community level.

## Implications for practice and/or policy

In developing countries including Ethiopia, low access to water supply, lack of adequate sanitation facilities, and low hygiene levels hinder proper MHM among reproductive-age women. A combination of inadequate WASH facilities, and poor MHM practices contribute to RTIs among reproductive-age women. Therefore, the findings of this study have implication for designing a women's health policy that focuses on availability of WASH facilities and health promotion in the improvement of MHM at the community level. This will require the development of integrated programs at the regional and community levels, where *kebeles* can play a major role in implementing these programs.

## Conclusions

The self-reported prevalence of RTI was 11.0% among reproductive-age women who were menstruating during the three months prior to data collection. We conclude that the following factors are associated with RTI: using unclean latrines/toilets, not washing hands with soap before touching the genital area, symptoms of RTI in the year prior to three months before the study period, having multiple sexual partners in the previous year, changing of blood-absorbent material only once per day during menstruation, and washing of the genital area only once per day during menstruation. Public health intervention measures need to consider these modifiable risk factors in order to reduce and prevent RTIs in Dessie City. Researchers should determine the prevalence of RTIs by including laboratory analysis to confirm RTI and identify the etiologic agents. Future research should examine the generalizability of these findings in other contexts and examine the causal relationships between sanitation infrastructure, hygiene practices, and women's health.

## Supporting information

**S1 File.**
(DOCX)

**S2 File.**
(DOCX)

## Acknowledgments

We acknowledge Dessie City Administration Health Office officials and health extension workers for their collaboration and valuable information. We also want to thank data collectors, supervisors, and study participants for their participation and valuable information.

## Author Contributions

**Conceptualization:** Ayechew Ademas, Metadel Adane, Tadesse Sisay, Betelhiem Eneyew, Awoke Keleb.

**Data curation:** Ayechew Ademas, Metadel Adane, Betelhiem Eneyew, Awoke Keleb, Mistir Lingerew.

**Formal analysis:** Ayechew Ademas, Metadel Adane, Tadesse Sisay, Betelhiem Eneyew, Awoke Keleb, Mistir Lingerew.

**Funding acquisition:** Ayechew Ademas, Metadel Adane.

**Investigation:** Ayechew Ademas, Metadel Adane, Tadesse Sisay, Betelhiem Eneyew, Awoke Keleb, Mistir Lingerew, Atimen Derso.

**Methodology:** Ayechew Ademas, Metadel Adane, Tadesse Sisay, Betelhiem Eneyew, Awoke Keleb, Atimen Derso, Kassahun Alemu.

**Project administration:** Ayechew Ademas, Metadel Adane, Tadesse Sisay, Mistir Lingerew.

**Resources:** Ayechew Ademas, Metadel Adane, Tadesse Sisay, Betelhiem Eneyew, Awoke Keleb, Mistir Lingerew, Atimen Derso, Kassahun Alemu.

**Software:** Ayechew Ademas, Metadel Adane, Tadesse Sisay, Awoke Keleb, Atimen Derso, Kassahun Alemu.

**Supervision:** Ayechew Ademas, Metadel Adane.

**Validation:** Ayechew Ademas, Metadel Adane, Helmut Kloos, Kassahun Alemu.

**Visualization:** Ayechew Ademas, Metadel Adane, Helmut Kloos, Kassahun Alemu.

**Writing – original draft:** Ayechew Ademas, Metadel Adane.

**Writing – review & editing:** Metadel Adane, Helmut Kloos.

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
