## [Decision Letter · Decision Letter 0]

23 Jul 2020

PONE-D-20-19169

Menstrual Hygiene Management, Water, Sanitation and Hygiene Remains a Public Health Problem for Women’s in Urbans of South Wollo, Amhara Region, Ethiopia

PLOS ONE

Dear Dr. Adane (PhD),

Thank you for submitting your manuscript to PLOS ONE. After careful consideration, we feel that it has merit but does not fully meet PLOS ONE’s publication criteria as it currently stands. Therefore, we invite you to submit a revised version of the manuscript that addresses the points raised during the review process.

We look forward to receiving your revised manuscript.

Kind regards,

Hans-Uwe Dahms, Ph.D.

Academic Editor

PLOS ONE

Journal Requirements:

3.We note that [Figure(s) 1] in your submission contain [map/satellite] images which may be copyrighted. All PLOS content is published under the Creative Commons Attribution License (CC BY 4.0), which means that the manuscript, images, and Supporting Information files will be freely available online, and any third party is permitted to access, download, copy, distribute, and use these materials in any way, even commercially, with proper attribution. For these reasons, we cannot publish previously copyrighted maps or satellite images created using proprietary data, such as Google software (Google Maps, Street View, and Earth). For more information, see our copyright guidelines: http://journals.plos.org/plosone/s/licenses-and-copyright.

1.    You may seek permission from the original copyright holder of Figure(s) [1] to publish the content specifically under the CC BY 4.0 license. 

Additional Editor Comments (if provided):

There are some issues that need to be addressed for a REVISED VERSION:

1- In Table 1 in the age section, 20-year-olds were divided into two parts (15 to 20 and 20 to 34 years old). Why?

2- What is the meaning of No formal education in Table 1? Do you mean illiterate people or people who have been educated at home?

3- What is meant by Above secondary? Does it mean university education? Or people with a PhD degree? Please write a note.

4- In Table 2 for Distance of latrine from the

house Written 15> and 15-30, Is 15 in the first category or the second?

Reviewers' comments:

Reviewer's Responses to Questions

**Comments to the Author**

1. Is the manuscript technically sound, and do the data support the conclusions?

Reviewer #1: Yes

Reviewer #2: Yes

2. Has the statistical analysis been performed appropriately and rigorously? 

Reviewer #1: Yes

Reviewer #2: Yes

3. Have the authors made all data underlying the findings in their manuscript fully available?

Reviewer #1: Yes

Reviewer #2: Yes

4. Is the manuscript presented in an intelligible fashion and written in standard English?

Reviewer #1: No

Reviewer #2: Yes

5. Review Comments to the Author

Reviewer #1: This article is valuable and can be published, but there are some issues that need to be addressed:

1- In Table 1 in the age section, 20-year-olds were divided into two parts (15 to 20 and 20 to 34 years old). Why?

2- What is the meaning of No formal education in Table 1? Do you mean illiterate people or people who have been educated at home?

3- What is meant by Above secondary? Does it mean university education? Or people with a PhD degree? Please write a note.

4- In Table 2 for Distance of latrine from the

house Written 15> and 15-30, Is 15 in the first category or the second?

Reviewer #2: This manuscript discusses a much needed work on Menstrual Hygiene Management, Water, Sanitation and Hygiene Remains a Public Health Problem for Women’s in Urbans of South Wollo, Amhara Region, Ethiopia with some basic results. The paper is suitable for publication with following corrections:

1. Add some of the most important quantitative results to the Abstract.

2. In Discussion and conclusion segment, authors can include the limitations of this study and more ideas and suggestions for prevention and control. The discussion section should be detailed with citation of all the recent references. Add more information about the significance and importance of this study.

3. A clear distinction of usage of the statistical tests has to be mentioned.

4. Authors should carefully review the final resolution of the figures prior to publication for better understanding of results.

5. Cite the source of future investigations in this study.

6. Rectify grammatical errors to reach up to international standards.

7. Ensure that the references and whole manuscript is as per journal format. Make certain that all the tables and figure citations are also in the standard format.

8.The authors are suggested to improve the quality of the language usage and correct grammatical errors.

6. PLOS authors have the option to publish the peer review history of their article (what does this mean?). If published, this will include your full peer review and any attached files.

Reviewer #1: No

Reviewer #2: No

---

## [Author Response · Author response to Decision Letter 0]

28 Jul 2020

Response to reviewers

Response: Thank you for this remark. We re-formatted the revised manuscript using the PLoS ONE format guidelines. The whole content of the manuscript, including the abstract, introduction, methods, discussion and reference are formatted using the guidelines (please see the revised version for each section).

Response: We provided the questionnaire in English version and original language as supporting information S I and S II, respectively. 

3. We note that [Figure(s) 1] in your submission contain [map/satellite] images which may be copyrighted. All PLOS content is published under the Creative Commons Attribution License (CC BY 4.0), which means that the manuscript, images, and Supporting Information files will be freely available online, and any third party is permitted to access, download, copy, distribute, and use these materials in any way, even commercially, with proper attribution. For these reasons, we cannot publish previously copyrighted maps or satellite images created using proprietary data, such as Google software (Google Maps, Street View, and Earth). For more information, see our copyright guidelines: http://journals.plos.org/plosone/s/licenses-and-copyright.

Response: The figure was removed in our submission. 

Reviewer #1

This article is valuable and can be published, but there are some issues that need to be addressed:

Response: Thank you for the positive comments on our paper. Your concerns are addressed below. 

1- In Table 1 in the age section, 20-year-olds were divided into two parts (15 to 20 and 20 to 34 years old). Why?

Response: Thank you very much for finding out such errors. The correct grouping is 15 to 19 (See Table 1). 

2- What is the meaning of No formal education in Table 1? Do you mean illiterate people or people who have been educated at home?

Response: Sorry for the confusion. We mean they are illiterate. Illiterate not attended both formal and informal education (See Table 1). 

3- What is meant by above secondary? Does it mean university education? Or people with a PhD degree? Please write a note

Response: Primary education means Grade 1-8; secondary education Grade 9-12, and above secondary education means that diploma, degree, masters and others. 

4- In Table 2 for Distance of latrine from the house Written 15> and 15-30, Is 15 in the first category or the second?

Response: 15 is the second category since for the first category we used less than 15. 

Reviewer #2

Reviewer #2: This manuscript discusses a much needed work on Menstrual Hygiene Management, Water, Sanitation and Hygiene Remains a Public Health Problem for Women’s in Urbans of South Wollo, Amhara Region, Ethiopia with some basic results. The paper is suitable for publication with following corrections:

Response: Thank you very much for the positive reflection of our paper. Corrections were made as follows. 

1. Add some of the most important quantitative results to the Abstract.

Response: We added some of the main quantitative results to the Abstract (See the result section of the abstract in lines 37 to 44. 

2. In discussion and conclusion segment, authors can include the limitations of this study and more ideas and suggestions for prevention and control. The discussion section should be detailed with citation of all the recent references. Add more information about the significance and importance of this study.

Response: Thank you for this pertinent comments. The discussion and conclusion section were revised (see the revised version in lines 363 to 367; 473-375; 379-382; 387-393; 397-399 and 416-419. See also about the conclusion in lines 462 to 464. The significance of the study in practice/policy implications were further explained (See lines 441 to 449.

3. A clear distinction of usage of the statistical tests has to be mentioned.

Response: The statistical testes are put in a clear way. See the revised lines Table 6. 

4. Authors should carefully review the final resolution of the figures prior to publication for better understanding of results.

Response: Figures 1 to 3 were deleted since they are not important. But Figure 4 during the revision became figure 1 and was updated for clarity (See figure 1). 

5. Cite the source of future investigations in this study.

Response: The recommendations for future investigations are the author’s idea and therefore need no source. 

6. Rectify grammatical errors to reach up to international standards.

Response: We edited the grammatical errors (see the revised version). 

7. Ensure that the references and whole manuscript is as per journal format. Make certain that all the tables and figure citations are also in the standard format.

Response: We formatted them as per the journal guidelines. 

8.The authors are suggested to improve the quality of the language usage and correct grammatical errors.

Response: We edited language usage and grammatical errors.

---

## [Editor Report · Decision Letter 1]

3 Aug 2020

Does menstrual hygiene management and water, sanitation, and hygiene predictors for reproductive tract infections among reproductive age Women in urban Areas of Ethiopia?

PONE-D-20-19169R1

Dear Dr. Adane (PhD),

We’re pleased to inform you that your manuscript has been judged scientifically suitable for publication and will be formally accepted for publication once it meets all outstanding technical requirements.

Kind regards,

Hans-Uwe Dahms, Ph.D.

Academic Editor

PLOS ONE

Additional Editor Comments (optional):

This MS has now be revised to an extend that it becomes acceptable.
---

## [Editor Report · Acceptance letter]

5 Aug 2020

PONE-D-20-19169R1 

Does menstrual hygiene management and water, sanitation, and hygiene predictors for reproductive tract infections among reproductive age Women in urban Areas of Ethiopia? 

Dear Dr. Adane (PhD):

I'm pleased to inform you that your manuscript has been deemed suitable for publication in PLOS ONE. Congratulations! Your manuscript is now with our production department. 

Kind regards, 

on behalf of

Dr. Hans-Uwe Dahms 

Academic Editor

PLOS ONE